# The Key Role of Intracellular 5-HT2A Receptors: A Turning Point in Psychedelic Research?

**Jacopo Sapienza** [1,2]

1   Department of Clinical Neurosciences, IRCCS San Raffaele Scientific Institute, 20127 Milan, Italy; sapienza.jacopo@hsr.it

2   Department of Humanities and Life Sciences, University School for Advanced Studies IUSS, 27100 Pavia, Italy

**Abstract:** Psychedelics could have revolutionary potential in psychiatry, although, until recently, the pharmacodynamic properties of such compounds have not seemed to differ much from those of serotonin, whose levels are raised by Serotonin Reuptake Inhibitors (SSRI). The cardinal point is that serotonergic compounds, such as antidepressive drugs, do not have the potential to induce long-lasting neuroplasticity as psychedelics do. Therefore, the biological underpinnings of the peculiar effect of such compounds had not been fully understood until new astonishing molecular findings came out this year to shed new light on them. Specifically, the phenomena of neuroplasticity are triggered by the stimulation of a peculiar type of receptors: the intracellular 5-HT2A receptors. Interestingly, psychedelics can reach this pool of intracellular receptors due to their lipophilic properties, as they can cross the lipophilic neuronal membrane while serotonin cannot. The importance of such a discovery should not be underestimated as the specific mechanisms involved have not yet been elucidated and a better understanding of them could pave the way to the development of new drugs (and/or new tailored therapeutic strategies) able to sustain neuroplasticity while minimizing side effects.

**Keywords:** serotonin; SSRI; neuroplasticity; synapses; cytoskeleton; dendrites; lipophilicity; endogenous psychedelics; DMT





## 1. Introduction

In the last decade, a revamped interest in psychedelic research spread in the scientific community as a "psychedelic renaissance" and mounting evidence on the potential applications of psychedelics in several psychiatric disorders was provided by clinical studies [1]. Nowadays, the antidepressant properties of psilocybin are widely reported by randomized clinical trials (RCTs) and confirmed by meta-analyses [2–6]. The evidence of efficacy is not limited to mood disorders, but also extends to the treatment of anxiety disorders, particularly in the context of life-threatening diseases [7,8] and substance use disorders [9,10], and several other trials are currently ongoing on in other clinical populations such as eating disorders, obsessive compulsive disorder, and neurocognitive disorders [11–15]. A plausible rationale also emerged for the treatment of schizophrenia, but possible implications are more controversial [16,17]. Although findings concerning the therapeutic role of such compounds being relatively new, the definition of psychedelics is still based on a classification dating back to the 1960s, and it is still under debate [18]. Classic (or serotonergic) psychedelics (semi-synthetics or derived from plants) are so called because their pharmacological effects are primarily mediated by the serotonergic system, as they are agonists or partial agonists of the 5-HT2A receptor [19,20]. Among the most used compounds in psychedelic research are Lysergic Acid Diethylamide (LSD), psilocybin and its active metabolite Psilocin, Mescaline and N,N-dimethyltryptamine (DMT). Concerning the serotonergic system, despite several attempts to uniquely clarify its role, its function remains elusive, constituting an enigma or a puzzle of too many pieces [21,22]. It has been

argued that such complexity may be due to its diversity in terms of receptor subtypes [23] and extensive innervation of the brain [24]. Interestingly, in vitro findings on the ability of psychedelics to induce neuroplasticity (neurite growth, sprouting of dendrites and synaptogenesis) shed a new light on other possible implications of the serotonergic system, and particularly of the 5-HT2A receptors [25–27].

## 2. 5-HT2A Receptors

The 5-HT2A receptor is one of at least 14 different 5-HT receptor subtypes in the mammalian brain [23], and like almost all of them, it is a G protein-coupled receptor (GPCR). The 5-HT2A receptor is the main excitatory GPCR of the serotonin receptor family, therefore the main effect of the 5-HT2A receptor is to increase the excitability of the neuron once serotonin binds to it [28]. The expression of 5-HT2A receptors is predominant in the cortex, being the most abundant serotonergic receptor in the cortical layers, particularly in high-level associative cortex [29]. Moreover, 5-HT2A receptors are mainly expressed on the dendrites of glutamatergic pyramidal neurons in layer V of the cortex [30] pointing at a putative mechanism of modulation elicited by them on such projecting neurons. The 5-HT2A receptors are known to be the primary target of antidepressants, particularly SSRI, thus this pool of receptors elicits antidepressant and anxiolytic properties when stimulated by increased levels of serotonin induced by antidepressant molecules [21].

## 3. The Historical Conceptual Issue

Since the discovery of the pharmacodynamic properties of LSD and other psychedelic compounds, astonishment and many questions have arisen in the scientific community regarding the overlap with serotonin or serotonergic compounds in terms of biological targets [22]. Although both classes of compounds showing a great affinity for the 5-HT2A receptors (5-HT2AR) acting as agonists, the clinical effects were totally different [31]. Psychedelics induce perceptual alterations such as illusion, visual hallucinations, hyperesthesia for colours, synaesthesia, change of meaning of perceptions, mystic experiences, alterations of ego boundaries such as ego dissolution, and altered consciousness, a sort of dreamlike state of consciousness [31,32]. Contrarily, serotonergic compounds or Selective Serotonin Reuptake Inhibitors (SSRI) do not elicit such effects and show antidepressant and anxiolytic properties [21]. It should be noted that there are a myriad of clinical trials on the antidepressant properties of psilocybin, and some anxiolytic effects have been reported as well [15]. As we are going to discuss later, such effects are probably the result of another underlying mechanism of action: neuroplasticity [25]. Nevertheless, a hallucinogenic effect has never been described among SSRI users. Why such discrepancies in terms of effects when the mechanism of action in terms of pharmacodynamics is the same? It has been argued that the main reason is the intracellular transduction pathway involved, which differs depending on the trigger at 5-HT2A receptors [25,33]. Although such a conclusion is plausible, it has not yet been confirmed by any findings and it remains unclear how this process could take place. It is important to note that until the specific mechanism of action of psychedelics is explained, it will not be possible to progress further in terms of tailored interventions or in the synthesis of molecules capable of minimizing unwanted/side effects and maximizing the potential of such compounds.

## 4. The Intracellular 5-HT2A Receptors

Although G protein-coupled receptors (GPCRs) such as 5-HT2A receptors are believed to be localized primarily to the plasma membrane, there is evidence proving also their intracellular localization [34–36]. Specifically, Cornea-Hébert et al. were among the first to describe a predominant cytoplasmatic localization of 5-HT2A receptors in rats using light and electron microscope immunocytochemistry and monoclonal antibodies against the N-terminal domain of the receptor protein [37]. A few years later, the same authors reported a predominant intracellular distribution of 5-HT2A receptors in the pyramidal neurons of the cerebral cortex, according to the distribution of 5-HT2A receptors located

transmembrane with the extracellular binding domain. In addition, they suggested a possible association of intracellular 5-HT2A receptors with the cytoskeletal microtubule-associated protein MAP1A [38]. The potential association with the cytoskeleton in cortical neurons is of great interest in light of more recent findings on the ability of psychedelics to modulate neuronal cytoarchitecture [25]. Indeed, Cornea-Hébert et al. hypothesised that such a pool of receptors could participate in intraneuronal signalling processes involved in cytoskeleton reorganization [38].

## 5. Psychedelics: A Matter of Lipophilicity

Over the years, unsolved questions concerning different putative mechanisms underlying the different effect of serotonergic compounds persisted, until February 2023 when Vargas et al. gathered many in vitro findings using molecular and genetic tools to show the underpinnings of psychedelic action [35]. The authors performed a series of experiments to clarify why some 5-HT2AR agonists (psychedelics) promote neuroplasticity, whereas others (serotonin) do not. To do so, they used DMT, d5-methoxy-N,N-dimethyltryptamine (5-MeO), Psilocin and related modified compounds (N-methylation) in order to decrease the polarity. They also used, in different sets of experiments, modified serotonergic non-psychedelic molecules in order to increase lipophilicity, or non-modified serotonergic compounds and ketanserin (and methylated ketanserin) with or without electroporation. The plasticity-promoting properties, and thus the ability to promote neurites growth, the sprouting of newborn dendrites and the arborisation of pre-existing ones, was measured as outcome (a fixed concentration of 10 μM was used for each compound) [35]. Indeed, there is much evidence pointing at neuroplasticity (rearrangements of cyto-architecture and creation of new synapses) as the pivotal mechanism of action of psychedelic compounds in a range of concentrations (in vitro) from 10 nM (LSD) to 10 μM [25,39]. In brief, Vergas et al. proved the involvement of a particular class of 5-HT2AR, the intracellular pool of 5-HT2AR, in the induction of neuroplasticity. The required, necessary and fundamental property of 5-HT2AR agonists in order to enhance neuroplasticity is lipophilicity, as membrane permeability is a requirement for psychedelic-induced neuroplasticity [35]. Interestingly, it was demonstrated that lipophilicity is directly proportional to psychoplastogenicity: the higher the lipophilicity, the higher the ability to induce neuroplasticity in target neurons. As a matter of fact, psychedelics can rearrange the conformation of neurons simply by crossing the lipophilic neuronal membrane and stimulating the intracellular pool of 5-HT2AR. On the other hand, polar compounds such as serotonin cannot do it because they cannot cross the membrane [35]. Overall, these results explain why serotonin and polar serotonergic compounds do not engage similar plasticity mechanisms: it is a matter of location.

## 6. Intracellular 5-HT2A Receptors: Possible Implications

### 6.1. Endogenous Ligands?

The location of 5-HT2AR is probably determinant in inducing different signalling cascades, and thus different effects especially in terms of neuroplasticity. The existence of such a pool of receptors with different characteristics compared to extracellular 5-HT2AR and the impossibility of serotonin to reach them could make the existence of specific ligands possible [35]. Interestingly, there is evidence of endogenous psychedelics synthesis in mammalians and even humans [40–42]. The N,N-dimethyltryptamine (DMT) is the result of the conversion of tryptamine, a metabolite of tryptophan, operated by indole-N-methyl transferase (INMT). The highest enzyme activity in the human brain is found in the subcortical layers of the fronto-parietal and temporal lobes and the cortical layers of the fronto-parietal lobe [40,41]. It could be hypothesized that this system, which encompass the intracellular 5-HT2AR pool and the endogenous DMT, sustains the phenomena of neuroplasticity in frontal and temporal areas by counteracting the physiological process of synaptic pruning, which becomes pathological in some psychiatric conditions.

*6.2. Therapeutic Implications: Personalized Interventions?*

The peculiarity of having two pools of receptors separated by a lipid membrane is that polar agonists can stimulate only the extracellular receptors, whereas non-polar or lipophilic compounds can bind both [35]. The possibility to induce the selective expression of intracellular 5-HT2AR in certain brain areas is fascinating because it should not produce any effects as serotonin cannot bind them until a non-polar agonist is administrated. Therefore, by selectively inducing an increased number of 5-HT2AR and then administering a psychedelic compound, enhanced neuroplasticity should be obtained only in the selected areas of the brain. Another strategy could be the selective inactivation of this pool of receptors except the target areas in order to achieve the abovementioned results. To do so, many current techniques allow specific molecules and drugs to be delivered to specific cell populations, even in the central nervous system (CNS) [43–47]. The main issue in the delivery of drugs in the CNS, in order to target specific neuronal populations, is represented by the blood–brain barrier (BBB) [44]. Focused ultrasound combined with intravenously injected microbubbles (FUS) transiently increase the permeability in specific regions of the BBB [46]. The concomitant administration of recombinant adeno-associated viruses (AAVs) intravenously allows them to cross the BBB at precise FUS-targeted brain regions [43]. Similarly, liposomes have been widely used in drug delivery in the CNS for the treatment and/or diagnosis of neurological diseases [44]. The covalent ligation of macromolecules as peptides, antibodies and RNA aptamers is an effective method for receptor-targeting liposomes, allowing their blood–brain barrier penetration and/or the delivery of therapeutic molecules to the target site [44,45]. Many other strategies could be hypothesized, but the cardinal point is the peculiarity of this two-steps process: (1) the number/availability/transduction/gene transcription/functionality of 5-HT2AR could be modulated in the first step of the process without any effect, because this class of receptor is inapproachable by non-psychedelic serotonergic compounds; (2) the subsequent administration of psychedelics will induce neuroplasticity depending on the topological pattern of 5-HT2AR availability previously induced.

*6.3. Do Lipophilic Properties Allow Storage?*

Some authors classified psychedelics as psychoplastogens. The compounds pertaining to this class show the ability to induce long-lasting neuroplasticity [46]. In other words, the neuroplastic effect persists beyond the period in which the molecule is circulating, and persistent growth of dendrites in the absence of psychedelic molecules in the extra-cellular space is a hallmark of serotonergic psychoplastogens [26,27]. Despite the growing knowledge base, it is still not currently clear how psychedelics can induce enduring neuroplasticity. Interestingly, an important piece was added to the puzzle recently with the discovery of the action of 5-HT2A receptors. It has been proposed that psychedelics could be stored in membranes of the Golgi apparatus as a substantial number of 5-HT2ARs in cortical neurons were found in the Golgi apparatus [35]. The compartments of such apparatus are slightly acidic compared with cytosol and extracellular space; the protonation of psychedelics in the Golgi apparatus could lead to the retention of psychedelics and the establishment of a sustained signalling versus the 5-HT2A receptors, which, in turn, results in persistent growth of neural terminals, even after transient stimulation [27,35].

## 7. Other Mechanisms of Action Involved in Psychedelic-Induced Neuroplasticity

Despite the crucial role played by the intracellular 5-HT2A receptors, the downstream mechanisms that lead to neuron growth and ramification have not been fully elucidated. They are likely to involve the Tropomyosin receptor kinase B (TrkB) and mammalian target of rapamycin (mTOR) signaling pathways, as previously demonstrated by several in vitro findings [25,39]. Another hypothesized mechanism of action is the psychedelic-induced increased levels of brain-derived neurotrophic factor (BDNF), which enhances AMPA receptor delivery to the synapse [48]. This mechanism, the AMPA receptor signaling, might be involved in the BDNF-mediated enhancement of neural plasticity elicited by

psychedelics [25,49]. Concerning other receptors that contribute to the complex mode of action of psychedelics, it has been demonstrated that LSD increases cortical spine density via 5-HT1A receptors stimulation. Precisely, such an effect might be due to 5-HT1A receptor desensitization [50]. Similar pharmacodynamic properties at the 5-HT1A receptor level were also reported for other psychedelics, such as psilocybin and DMT, suggesting possible similar mechanisms of action [51]. Interestingly, the formation of heterodimeric membrane receptor complexes (e.g., D2-5-HT2A) and the stimulation elicited on them by psychedelics might trigger different intracellular pathways, further contributing of the complexity of such compounds in terms of mechanism of action [52,53].

## 8. Conclusions

The discovery of the importance of 5-HT2A receptors provides a new viewpoint on the mechanism of action of psychedelics and could be a milestone in psychedelic research, hopefully in terms of personalized interventions or the synthesis of new therapeutic compounds with different pharmacological properties and effects. The intracellular location and the indirect interactions with the cytoskeleton probably facilitate the remodelling of neuronal cytoarchitecture, the sprouting of dendrites and, overall, the arborization of neurons [38,54]. On the other hand, it is possible that the chemical properties of psychedelics allow intracellular retention and gradual intracellular diffusion towards the receptors, thus causing continuous stimulation which, in turn, sustains the long-lasting neuroplasticity [35]. All these cues should be considered for future hypotheses in the field of drug development.

**Funding:** This research received no external funding.

**Conflicts of Interest:** The author declares no conflict of interest.

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
