# Peer review of "The Key Role of Intracellular 5-HT2A Receptors: A Turning Point in Psychedelic Research?"

_psychoactives, doi:10.3390/psychoactives2040018_

Round 1

Reviewer 1 Report

Following the work recently published by Vargas et al, Sapienza presents a brief but clear assessment of the therapeutic implications of intracellular 5HT2A receptors identification of as a potential target of psychedelics.

Although the evidence of Vergas et al., is solid, a too relevant part of the text is focused on that paper.  The author should also mention other mechanisms potentially involved in the induction of neuroplasticity by psychedelics. For example, an important field of research is the possible role of heterodimeric membrane receptor complexes that allow the activation of distinct downstream signaling pathways.

Moreover, although the intracellular 5-HT2A receptor could represent the major psychedelics target, the author should point out the possible role of other molecular targets or mechanism of action, as often reported for antipsychotics, and psychoactive drugs in general, with lipophilic and/or cationic amphiphilic properties.

These evaluations should be made in order to draw attention to the complex mode of action of these compounds which underlies the multiplicity of effects, some of which, undesired in psychiatric disorders therapy.

Conclusions should be adjusted.  If the author is of the opinion that space should be given to molecular pathways downstream 5HT2A - and it is a more than reasonable choice - a brief dissertation on the molecular mechanisms behind the neuroplasticity induced by psychedelics will require an independent paragraph. 

Author Response

I thank the reviewer for the valuable comments, I agree with him. I created the paragraph 7 to discuss the other main findings on the possible mechanisms underlying psychedelic-induced neuroplasticity. They range from the intracellular pathways of mTOR and TrkB to heterodimeric membranereceptor complexes and the role of 5HT1AR (as suggested also by reviewer 3), AMPAR and BDNF. I also reshaped the conclusion section accordingly. 

Reviewer 2 Report

Author presented the recent state of research about psychedelics starting from historical issues through mechanism and therapeutic efficacy. The most important issue is discovery of intracellular 5-HT2A receptors, which is a milestone in psychedelic research opening the possibility of new molecules development in the absence of side effects.

In paragraph 3 - the abbreviation for 5-HT2A receptors (5HT2AR) is not necessary if it is not used in the rest of the text.

Author Response

Thank you. I apologize for the oversight. I corrected the mistake.

Reviewer 3 Report

The manuscript “Intracellular 5HT2A receptors: a milestone in psychedelic research?" by J Sapienza reviews the role of the activation of the 5-HT2A receptor by psychedelics as a key event in these substances’ ability to enhance neuroplasticity. The manuscript is well-structured, but there are a few issues that need to be addressed before it can be accepted for publication. These are detailed below:

1.   The title needs to be rephrased. Usually, a milestone comprises an event or an action. In that sense, “Intracellular 5HT2A receptors” do not seem to be a milestone.

2. It would be important for readers outside of this specific field of research who may come in contact with this manuscript that the author clearly provides some examples of the psychedelics he refers to.

3. It would be interesting to discuss the potential allosteric binding of psychedelics to the 5HT2A receptors, as well as the role of the binding of psychedelics to other receptors, like the 5-HT1A in the neuroplasticity-promoting properties of these substances.

4. It would also be interesting to mention the range of concentrations/doses at which psychedelics promote neuroplasticity.

Minor comments

1.      Line 16: you may use only “psychedelics” instead of “psychedelics compounds” 

2.      I am not sure whether it is a formatting guideline from the journal, but, usually, consecutive in-text citations are separated by a hyphen (e.g., [2-6] instead of [2][3][4][5][6]).

3.      Line 64: Please indicate the name of LSD in full (although it is a well-known molecule, readers from different fields of research may not understand it).

4. Line 79: I believe the author meant to write “when” instead of “whether”.

There are several typos and syntax errors throughout the manuscript that need to be corrected.

Author Response

1) I thank the reviewer for the comment. I reformulated the title.

2) Again, I agree with the reviewer. I provided the most common and used molecules in psychedelic research in the introduction. Moreover, I also reported the molecules that Vergas et al. used to perform their experiments in the paragraph 5.

3) As suggested also by reviewer 1 I created an additional paragraph to discuss other mechanisms involved in neuroplasticity in whihch I included also the evidence on the 5HT1AR.

4) I thank the reviewer for the comment. I provided the concentrations used in the series of experiments by Vergas et al. and in other in vitro experiments. However, to assess the oral dose that corresponds to an enhancement of neuroplasticity in humans is difficult as it is not feseable in vivo. Changes in connectivity (fRMI) or the subjective hallucinogenic effect could be measured but they are not direct evidence.

I thank the reviewer also for Minor comments:

1) Done

2) It is true, I modified the manuscript accordingly.

3) Done

4)  I apologize, yes I corrected the mistake

Typos and mistakes were corrected

Round 2

Reviewer 1 Report

the revisions are adequate. I have no other comments

Reviewer 3 Report

The author has properly addressed all my comments.